**Data Availability Statement:** All relevant data are within the paper and its Supporting information files.

# Round-window delivery of lithium chloride regenerates cochlear synapses damaged by noise-induced excitotoxic trauma via inhibition of the NMDA receptor in the rat

Ji Eun Choi[1,2☯], Nathaniel T. Carpena[2☯], Jae-Hun Lee[3], So-Young Chang[2], Min Young Lee[1,2], Jae Yun Jung[1,2‡]*, Won-Ho Chung[4‡]*

**1** Department of Otolaryngology Head and Neck Surgery, Dankook University Hospital, College of Medicine, Dankook University, Cheonan, South Korea, **2** Multi-modality Treatment Research Center for Auditory/Vestibular Disease, College of Medicine, Dankook University, Cheonan, South Korea, **3** Center for Cognition and Sociality, Institute for Basic Science, Daejeon, Republic of Korea, **4** Department of Otorhinolaryngology-Head and Neck Surgery, Samsung Medical Center, Sungkyunkwan University School of Medicine, Seoul, Korea

☯ These authors contributed equally to this work.
‡ JYJ and W-HC also contributed equally to this work.
* jjkingy2k@gmail.com (JYJ); whchung@skku.edu (WHC)

## Abstract

Noise exposure can destroy the synaptic connections between hair cells and auditory nerve fibers without damaging the hair cells, and this synaptic loss could contribute to difficult hearing in noisy environments. In this study, we investigated whether delivering lithium chloride to the round-window can regenerate synaptic loss of cochlea after acoustic overexposure. Our rat animal model of noise-induced cochlear synaptopathy caused about 50% loss of synapses in the cochlear basal region without damaging hair cells. We locally delivered a single treatment of poloxamer 407 (vehicle) containing lithium chloride (either 1 mM or 2 mM) to the round-window niche 24 hours after noise exposure. Controls included animals exposed to noise who received only the vehicle. Auditory brainstem responses were measured 3 days, 1 week, and 2 weeks post-exposure treatment, and cochleas were harvested 1 week and 2 weeks post-exposure treatment for histological analysis. As documented by confocal microscopy of immunostained ribbon synapses, local delivery of 2 mM lithium chloride produced synaptic regeneration coupled with corresponding functional recovery, as seen in the suprathreshold amplitude of auditory brainstem response wave 1. Western blot analyses revealed that 2 mM lithium chloride suppressed N-methyl-D-aspartate (NMDA) receptor expression 7 days after noise-exposure. Thus, round-window delivery of lithium chloride using poloxamer 407 reduces cochlear synaptic loss after acoustic overexposure by inhibiting NMDA receptor activity in rat model.

**Funding:** This work was supported by a National Research Foundation of Korea (NRF) grant funded by the Korean government (MSIT) to JEC [no. NRF-2020R1C1C1009849] and by the Basic Science Research Program through the NRF founded by the Ministry of Education in the form of funding to JEC [NRF-2020R1A6A1A03043283, RS-2023-00208177]. The funders had no role in study design, data collection and analysis, decision to publish, or preparation of the manuscript.

**Competing interests:** The authors have declared that no competing interests exist.

## Introduction

Noise can cause temporary or permanent damage to the inner ear depending on exposure characteristics such as the sound pressure level (SPL), exposure time, and noise spectrum, as well as characteristics of the individual [1]. Exposure to loud sounds can destroy cochlear sensory hair cells, causing permanent elevation of the hearing threshold [2–4]. Even at an SPL too low to permanently damage cochlear hair cells, noise exposure can impair hearing by destroying the cochlear ribbon synapses between inner hair cells (IHCs) and spiral ganglion neurons (SGNs) [5, 6]. Although this cochlear synaptopathy does not elevate the threshold, it could contribute to difficulties in understanding speech in noisy environments, tinnitus, and hyperacusis [7, 8].

The accumulated evidence indicates that noise can cause excitotoxic trauma to cochlear synapses by triggering the excessive release of the neurotransmitter glutamate from auditory hair cells [9–12]. Direct application of glutamate agonist causes swelling of the SGN terminals in regions of synaptic contact with IHCs, similar to the morphological changes seen after acoustic overexposure [9, 13–15]. Conversely, a glutamate receptor blockade reduces swelling of the post-synaptic terminals during acoustic overexposure [10–12]. Although the downstream mechanisms of noise-induced damage are unknown, those observations suggest that the primary cause of noise-induced synaptic loss is the activation of glutamate receptors.

Recently, lithium has been reported to show neuroprotective effects against glutamate excitotoxicity, and that neuroprotective action has been associated with the inactivation of *N*-methyl-D-aspartate (NMDA) receptors [16–18]. Increasing evidence supports the notion that lithium-induced inhibition of phosphorylation of the NMDA receptor's NR2B subunit is likely to result in its inactivation and contribute to neuroprotection against excitotoxicity [18, 19]. Prior studies have shown that lithium can protect against cisplatin-induced cytotoxicity in auditory cells [20] and attenuate aging-related auditory cortex apoptosis [21]. However, there is a lack of evidence about whether lithium can reverse noise-induced cochlear synaptopathy.

Our aim in the present study was to evaluate whether lithium has neuroprotective effects in noise-induced cochlear synaptopathy. We therefore used a thermoreversible hydrogel to deliver lithium directly to the round-window in a rat model of noise-induced cochlear synaptopathy and conducted functional and histological analyses to investigate whether lithium reduced synaptic loss after acoustic overexposure. In addition, using western blot analyses, we explored the role of changes in the NMDA receptor in mediating neuroprotection in noise-induced cochlear synaptopathy.

## Methods

### Animals

Male Sprague-Dawley rats (5 weeks old; 130–150 g body weight; Nara Biotech, South Korea) were used for this study. The animals were housed in the laboratory animal facility and given free access to food and water. Before starting the experimental procedures, all rats were acclimated to their housing conditions for 1 week and measured the baseline auditory brainstem response (ABR) threshold.

Based on the ABR measurement, forty rats with normal hearing threshold were arbitrarily assigned to one of five groups. The group 1 was not exposed to noise or surgery to assess normal cochlear synaptic counts (normal group, n = 8). The group 2 was exposed to a noise band designed to destroy cochlear synapses (noise-only group, n = 4). The three other groups underwent surgery 24 hours after noise exposure to place a solution containing only vehicle (Group 3: noise + vehicle, n = 14), 1 mM lithium chloride (LiCl; Group 4: noise + 1 mM LiCl,

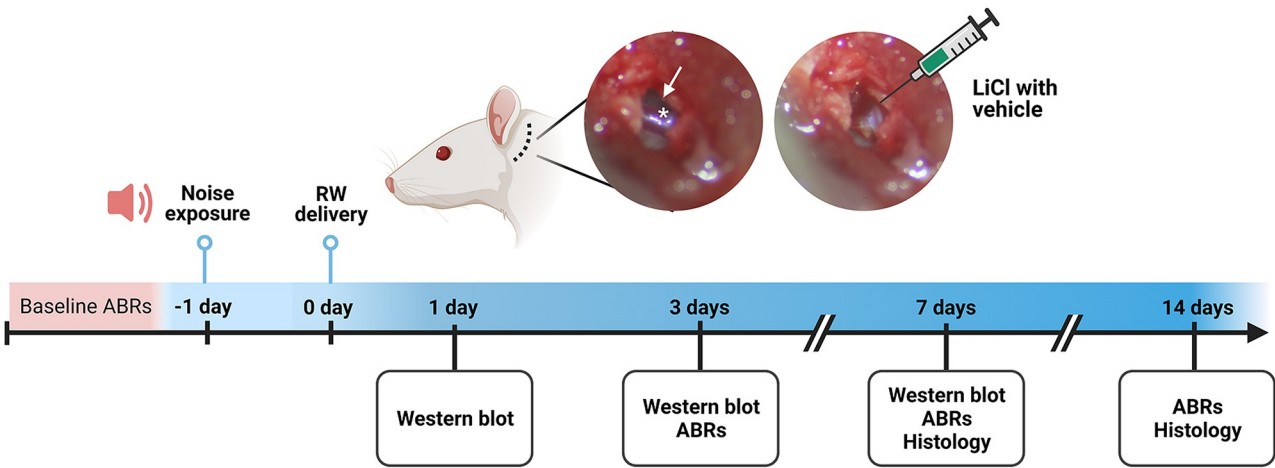

**Fig 1. The otic bulla was exposed via postauricular incision, and a tiny hole was made to visualize the round-window membrane (arrow) and stapedial artery (asterisk).** Poloxamer 407 (P407) solution with or without LiCl was delivered to the round-window. ABR; auditory brainstem response, RW; round-window, WB; western blot.

n = 10), or 2 mM LiCl (Group 5: noise + 2 mM LiCl, n = 14), onto the round-window membrane.

Fig 1 shows a schematic timeline of the experimental protocol. All experimentation was carried out in strict accordance with the recommendations in the *Guide for the Care and Use of Laboratory Animals* of the National Institutes of Health. The protocol was approved by the Institutional Animal Care and Use Committee of the University of Dankook (Protocol Number: DKU-17-041 and DKU-19-040). After the completion of all experiments, animals were euthanized by tiletamine–zolazepam/xylazine overdose followed by cervical dislocation.

## Noise exposure

The awake and unrestrained animals were exposed to a narrow band of noise (16 kHz with 1 kHz of bandwidth) for 2 hours at an SPL of 105 dB. During that time, the animals were placed in individual cages to prevent defensive behaviors, and those cages were placed in custom-made soundproof acryl chambers equipped with a BEYMA CP800Ti speaker (Beyma, Valencia, Spain). The noise was generated with a type 1027 sine random generator (Bruel and Kjaer, Denmark) and amplified with an R300 plus amplifier (Inter-M, Seoul, Korea). The sound levels were verified in the floor of the chamber using a frequency-specific sound level meter (Sound Level Meter-Type 2250; Bruel and Kjaer, Copenhagen, Denmark) before and during each noise exposure.

## Round-window delivery of LiCl

An 18% (w/v) stock solution was prepared by slowly dissolving poloxamer 407 (Sigma-Aldrich Cat# 16758) in sterile water. LiCl powder (42.39 g/mol) (Sigma-Aldrich Cat# L7026) was reconstituted in the 18% poloxamer 407 solution, yielding a final concentration of 1 mM as the low dose and 2 mM as the high dose, and stored overnight in a refrigerator at 4°C to ensure complete dissolution. The 18% poloxamer 407 solution is liquid in the refrigerator and room temperature but becomes highly viscous at body temperature.

The animals were anesthetized by an intraperitoneal injection of tiletamine–zolazepam (30 mg/kg) and xylazine (10 mg/kg) and positioned right ear-down. A postauricular skin incision

was made, and the subcutaneous tissues and superficial fascia of the neck were bluntly dissected. After exposing the otic bulla, a tiny hole was made and enlarged until the round-window membrane was clearly visible (Fig 1). A 16G needle was positioned within the round-window niche, and 50 μl of the poloxamer solution was injected using a 1 mL syringe over the round-window membrane. The hole was sealed with muscle, and the wound was closed with 4–0 Vicryl sutures (Ethicon). Intramuscular injection of ketamine (10mg/kg) was applied after surgery (prior to anesthetic recovery) to minimize animal suffering.

## Auditory brainstem response measurements

ABRs were measured using a Davis Technologies system (TDT system III, Alchua, Florida, USA) to identify the degrees of hearing loss and recovery. The animals were anesthetized by an intraperitoneal injection of tiletamine–zolazepam (30 mg/kg) and xylazine (10 mg/kg) and placed in a sound-proof chamber. Five millisecond tone-burst stimuli (0.5 ms cos2 rise-fall) at four frequencies (8, 12, 16, and 32 kHz) were delivered with alternating polarity. Electrical responses were collected via needle electrodes at the vertex (active) and ventral edge of each pinna (reference and ground), amplified 10,000X with a 0.3–3 kHz passband, and averaged using 1,024 responses at each SPL. The responses were collected for stimulus levels in 5 dB steps from 90 to 10 dB SPL. The ABR threshold was defined as the lowest sound level at which a reproducible waveform could be observed. The wave 1 amplitude was defined as the difference between the first positive peak and the following negative peak. Hearing thresholds and wave 1 amplitudes were obtained before noise exposure (baseline) and 3, 7, and 14 days after the treatment.

## Cochlear processing and immunohistochemistry

Following the endpoint ABR measurements, thirty-eight animals were sacrificed (n = 2 for group 1 and 2, n = 5 for group 3–5 at 1 week and 2 weeks post-treatment). After anesthesia with intraperitoneal tiletamine/zolazepam (30 mg/kg)–xylazine (10 mg/kg), their cochleae were quickly harvested. The harvested cochleae were fixed in 4% paraformaldehyde overnight. After being washed with 0.1 M phosphate buffered saline (PBS), the cochleae were decalcified in 0.5 M EDTA (pH 8.0) and micro-dissected into 4–5 pieces for whole-mount cochlear processing. For immunostaining, the cochlear pieces were blocked with 5% normal goat serum (NGS) in PBS and 0.3% Triton X-100 (TX) at room temperature for 1 hour followed by overnight incubation at 4˚C with the following primary antibodies diluted in 1% NGS with 0.3% TX: 1) mouse (IgG1) anti-CtBP2 (C-terminal binding protein) at 1:500 (BD Transduction Labs Cat# 612044, RRID: AB_399431) to quantify pre-synaptic ribbons and 2) rabbit anti-myosin 7a at 1:200 (Proteus Biosciences Cat# 25–6790, RRID: AB_10015251) to delineate hair cells. After being washed three times with PBS for 5 min, the cochlear pieces were incubated for 1 hour at room temperature in species-appropriate secondary antibodies: 1) Alexa Fluor 488-conjugated goat anti-mouse (IgG2a) at 1:1000 (Molecular Probes Cat# A-21131, RRID: AB_141618) or 2) Alexa Fluor 568-conjugated chicken anti-rabbit at 1:200 (Innovative Research Cat# A21443, RRID: AB_1500685). After being washed three times with PBS for 5 min, the stained cochlear pieces were slide-mounted using Vectashield (Vector Labs) and cover-slipped.

## Synapse and hair cell counts

Images were obtained using a confocal microscope (Flow-View 3000, Olympus, Japan) with a glycerol-immersion 40X objective and 2X digital zoom. For each image stack, the z dimension was sampled at 25 μm with a resolution of 0.5 μm per slice, and the span was adjusted to

include all synaptic elements in the xy field of view. One z-stack was obtained in each frequency location, and each x-stack included 8–12 adjacent IHCs. To identify and count hair cell synapses, image stacks were imported into image processing software (ImageJ ver. 1.43u, NIH, US). The outer hair cells (OHCs) and IHCs in each stack were counted using faint nuclear staining by anti-CtBP2 and hair-cell staining by anti-myosin 7a. To quantitatively assess pre-synaptic ribbons, pre-synaptic markers within 10–11 IHCs were counted and divided by the number of IHCs. The location of the basilar membrane at the relevant frequency was determined by computing a cochlear frequency map [22].

## Western blot analysis

To perform the western blot, cochlear tissues were additionally dissected under the anesthesia with intraperitoneal tiletamine/zolazepam (30 mg/kg)–xylazine (10 mg/kg) from group 1 (normal), group 3 (noise + vehicle), and group 5 (noise + 2 mM LiCl). Four cochlear tissues per group were pooled with 3 replicates and then homogenized in radioimmunoprecipitation assay buffer (Biosesang, Seongnam, Gyeonggi, Korea) containing protease (1:100, Sigma-Aldrich Cat# P8340) and phosphatase inhibitor cocktails (1:100, Sigma-Aldrich Cat# P8340). The cochlear lysates were centrifuged at 12,000 rpm and 4˚C for 15 min. The collected supernatants were collected and stored at -20˚C until further use. Equal amounts of proteins were mixed with Laemmli loading buffer (Bio-Rad Cat# 161–0737) and β-mercaptoethanol (0.7 mM), boiled at 95˚C for 5 min, and then separated using 6% SDS-polyacrylamide gel. After electrophoresis, the gel proteins were electro-transferred onto an immunoblot nitrocellulose membrane (Bio-Rad Laboratories, CA, USA). The membranes were blocked with 5% skimmed milk and incubated overnight at 4˚C with primary antibodies to anti-NR2B (1:500; Abcam Cat# ab28373, RRID: AB_776810), anti-phospho-NR2B (Ser1303) (1:500; Abcam Cat# ab81271, RRID: AB_1640731), or anti-β-actin (1:2,000; Abcam Cat# ab6276, RRID: AB_2223210). After being washed with PBS and Tween-20 (PBST; 0.1M PBS, 0.1% Tween-20), the membranes were incubated with horseradish peroxidase–conjugated goat anti-rabbit (1:2000; Thermo Fisher Scientific Cat# A16104, RRID: AB_2534776) or goat anti-mouse secondary antibody (1:2000; Thermo Fisher Scientific Cat# A16072, RRID: AB_2534745) in PBS with 2.5% skimmed milk. After being thoroughly washed with PBST, the membranes were visualized by enhancing chemiluminescence substrate (Pierce, Rockford, IL) for 2 min, followed by chemiluminescence detection on ChemiDoc XRS+ System (Bio-Rad Laboratories, Hercules, CA, USA). The band intensities were quantified using image processing software (ImageJ ver. 1.43u, NIH, Bethesda, MD, US), and the relative expression of each protein was normalized to β-actin. Fold change was calculated by comparing to the normal group.

## Statistical analysis

Statistical analyses were conducted using Prism software 8.4.3 (GraphPad Software, La Jolla, CA, USA, RRID:SCR_002798). All datasets were tested for normal distribution using the Shapiro-Wilk test and QQ-plots. Outlier were identified by ROUT test and dataset had no outliers. The ABR thresholds, ABR wave 1 amplitude, synapse counts, and western blot result were compared between experimental groups (Group 3–5). For multiple groups, nonparametric data were statistically analyzed using the Kruskal-Wallis H test and the two-way repeated measures analysis of variance (ANOVA). If there were significant differences among the experimental groups, post-hoc analysis with Dunn's multiple comparisons test was performed to evaluate differences between two groups. Tests were considered statistically significant when $p$-value and adjusted $p$-value were less than 0.05.

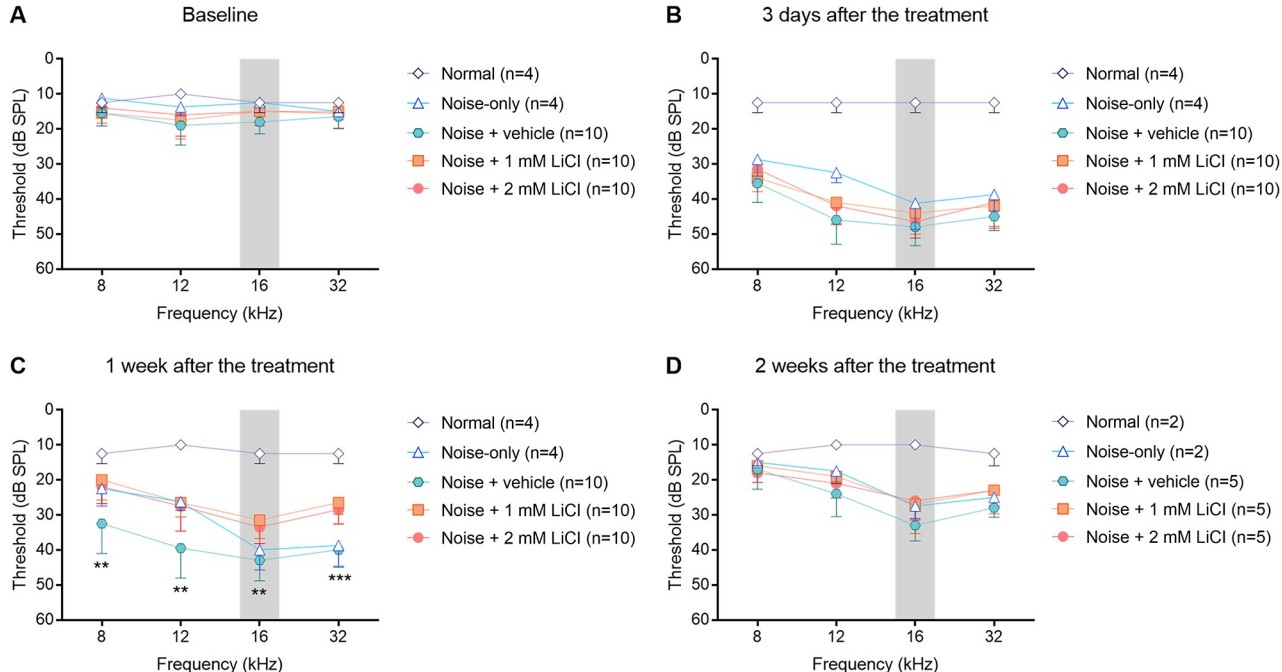

**Fig 2. Mean plots of ABR thresholds measured at baseline (A), 3 days (B), one week (C), and two weeks (D) after the treatment.** Noise-induced threshold shifts one week after exposure differed significantly among the experimental groups (noise + vehicle, noise + 1 mM LiCl, and noise + 2 mM LiCl). The gray shaded area indicates narrow-band noise spectrum. Error bars show standard deviation. The statistical significance of group differences is indicated by asterisks: * $p < 0.05$, ** $p < 0.01$, *** $p < 0.001$ based on the Kruskal-Wallis H test. n = number of animals, ABR; auditory brainstem response, dB SPL; decibel sound pressure level.

## Results

### Early recovery of temporary noise-induced threshold shifts after round-window delivery of LiCl

ABRs were tested to examine the threshold of hearing before and after the noise exposure and round-window drug delivery. As observed by comparing the normal and noise-only groups, noise exposure caused a threshold shift of ~30 dB SPL at frequencies above the noise band, when measured 1 week after the treatment, (Fig 2). Two week later, ABR thresholds recovered up to 15 dB SPL. Finally, noise exposure caused a permanent ~15 dB SPL threshold shift at frequencies above the noise band. Compared to the noise-only group, the noise + vehicle group had worse ABR threshold at 8 kHz and 12 kHz at 1 week after the treatment, but ABR threshold in the noise + vehicle group had returned to that of the noise-only group.

The delivery of LiCl to the round-window produced early recovery of the temporary noise-induced threshold shift (Fig 2b). The Kruskal-Wallis H test showed significant threshold differences among the experimental groups (noise + vehicle, noise + 1 mM LiCl, and noise + 2 mM LiCl groups) one week after the treatment (8 kHz: $\chi^2(2) = 11.859$, $p = 0.003$; 12 kHz: $\chi^2(2) = 12.071$, $p = 0.002$; 16 kHz: $\chi^2(2) = 14.256$, $p = 0.001$; 32 kHz: $\chi^2(2) = 19.090$, $p < 0.001$). The post-hoc analysis with Dunn's multiple comparisons test revealed that the hearing thresholds recovered earlier in the noise + 1 mM LiCl (8 kHz: $p = 0.003$, 12 kHz: $p = 0.006$, 16 kHz: $p = 0.001$, 32 kHz: $p < 0.001$) and noise + 2 mM LiCl (8 kHz: $p = 0.031$, 12 kHz: $p = 0.010$, 16 kHz: $p = 0.009$, 32 kHz; $p = 0.002$) groups than in the noise + vehicle group. However, there were no significant differences among the experimental groups two weeks after the treatment

(8 kHz: $\chi^2(2)$ = 1.056, $p$ = 0.590; 12 kHz: $\chi^2(2)$ = 2.491, $p$ = 0.288; 16 kHz: $\chi^2(2)$ = 3.583, $p$ = 0.167; 32 kHz: $\chi^2(2)$ = 4.154, $p$ = 0.125).

## Functional recovery of suprathreshold responses after round-window delivery of LiCl

Changes in the mean peak 1 amplitude after noise exposure are shown in Fig 3. The mean peak 1 amplitudes upon suprathreshold stimuli (all levels from 65–80 dB SPL) were reduced at frequencies above the noise band, where the maximal noise-induced threshold shift was seen. All experimental groups had statistically significant differences before and after noise exposure as determined by two-way ANOVA for all levels from 65–80 dB SPL (noise + vehicle group: F(3, 27) = 28.15, $p$ < 0.001 for 16 kHz and F(3, 27) = 40.1, $p$ < 0.001 for 32 kHz; noise + 1 mM LiCl: F(3, 27) = 41.4, $p$ < 0.001 for 16 kHz and F(3, 27) = 24.3, $p$ < 0.001 for 32 kHz; noise + 2 mM LiCl: F(3, 27) = 22.67, $p$ < 0.001 for 16 kHz and F(1.674, 15.07) = 23.58, $p$ < 0.001 for 32 kHz).

Two weeks after the treatment, the mean peak 1 amplitudes had not completely recovered in the noise + vehicle (all adjusted $p$ < 0.001 by post-hoc analysis with Dunnett's multiple comparisons test except for stimulus levels from 70- and 65-dB SPL at 16 Hz) and noise + 1 mM LiCl groups (all adjusted $p$ < 0.001 by post-hoc analysis with Dunnett's multiple comparisons test). However, the suprathreshold amplitudes of the ABRs had fully recovered in the noise + 2 mM LiCl group two weeks after the treatment. The suprathreshold peak 1 amplitude at 16 kHz and 32 kHz did not differ significantly between baseline and two weeks post-exposure treatment in the noise + 2 mM LiCl group (all adjusted $p$ > 0.05 by post-hoc analysis with Dunnett's multiple comparisons test).

## Synaptic regeneration after round-window delivery of LiCl

Cochlear tissue immunostained with antibodies specific for hair cells (myosin 7a) and ribbon synapses (CtBP2) showed no loss of OHCs or IHCs; instead, there was a loss of IHC synapses after noise exposure (Fig 4). The loss of IHC synapses is clearly visible in the immunostained cochlear tissue from the noise + vehicle group compared with the normal group (Fig 4A and 4B). As quantitatively shown in Fig 4C and 4D, the mean synaptic count per IHC was reduced in the noise-exposed groups (noise-only and noise + vehicle groups), mainly in the frequencies above the noise band (16 kHz and 32 kHz).

Synaptic rescue can be seen in the noise + 2 mM LiCl group both one week (Fig 4A) and two weeks (Fig 4B) after the treatment. The Kruskal-Wallis H test showed a statistically significant difference among the experimental groups (noise + vehicle, noise + 1 mM LiCl, noise + 2 mM LiCl groups) in the mean pre-synaptic counts at 16 kHz and 32 kHz one week (16 kHz: $\chi^2(3)$ = 10.50, $p$ < 0.001; 32 kHz: $\chi^2(3)$ = 12.50, $p$ < 0.001) and two weeks (16 kHz: $\chi^2(3)$ = 9.637, $p$ = 0.002; 32 kHz: $\chi^2(3)$ = 11.18, $p$ < 0.001) after the treatment. The post-hoc analysis with Dunn's multiple comparisons test revealed that the mean pre-synaptic counts in the noise + 2 mM LiCl group were significantly higher than those in the noise + vehicle group (all adjusted $p$ < 0.5 for 16 kHz and 32 kHz when measured one week and two weeks after the treatment). However, the mean pre-synaptic counts did not differ significantly between the noise + vehicle and noise + 1 mM LiCl groups.

## Neuroprotective effects of round-window delivery of LiCl against glutamate excitotoxicity

Fig 5 shows representative western blots and corresponding quantitative analyses of total NR2B and phospho-NR2B (pNR2B) expression. The Kruskal-Wallis test was conducted to

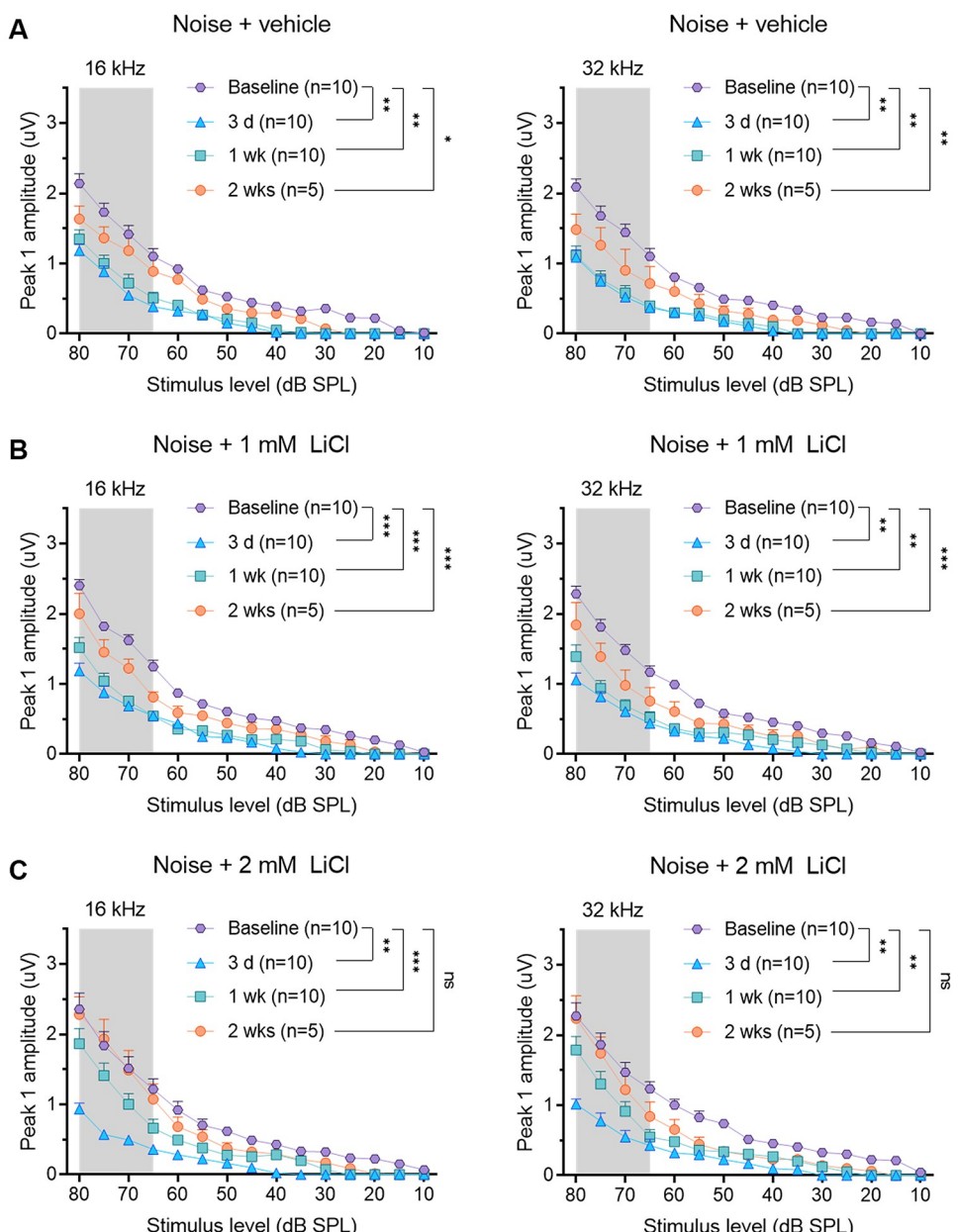

**Fig 3. Mean amplitude vs. level functions (± SEM) of ABR wave 1 at frequencies above the noise band measured in the noise + vehicle (A), noise + 1 mM LiCl (B), and noise + 2 mM LiCl (C) groups.** The gray shaded area indicates suprathreshold stimuli (all levels from 65–80 dB SPL). Statistically significant differences before and after noise exposure are shown by asterisks: * adjusted $p < 0.05$, ** adjusted $p < 0.01$, *** adjusted $p < 0.001$ based on a post-hoc analysis with Dunnett's multiple comparisons test for all levels from 65–80 dB SPL. n = number of animals, ABR; auditory brainstem response, dB SPL; decibel sound pressure level.

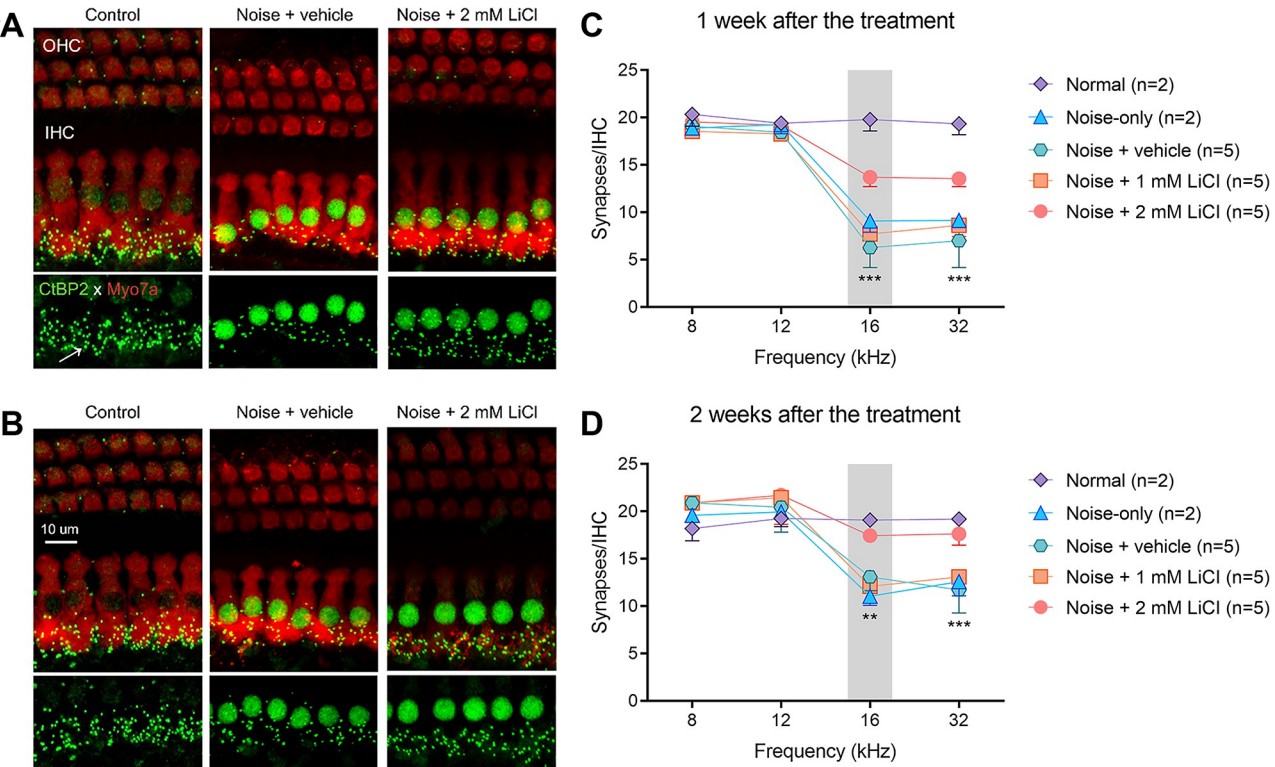

**Fig 4. High-dose LiCl delivery can rescue the noise-induced loss of inner hair synapses. A, B** Cochlear tissue immunostained with specific antibodies for hair cells (myosin 7a, red) and ribbon synapses (CtBP2, green) in the 16 kHz region one week (**A**) and two weeks (**B**) after noise exposure. The arrow indicates the position of a pre-synapse. **C, D** Quantification of the number of ribbon synapses at the relevant frequencies of 8 kHz, 12 kHz, 16 kHz, and 32 kHz. Median plot of pre-synaptic count was compared among the experimental groups (noise + vehicle, noise + 1 mM LiCl, and noise + 2 mM LiCl). Error bars show 95% confidence intervals. Statistical significance is indicated by asterisks: * $p < 0.05$, ** $p < 0.01$, *** $p < 0.001$ based on the Kruskal Wallis test. n = number of animals.

compare the fold changes of total NR2B and pNR2B levels among the normal, noise + vehicle, and noise + 2 ml LiCl groups on days 1, 3, and 7 after the treatment. The levels of total NR2B and pNR2B were unchanged until 3 days after the noise exposure and/or the treatment [NR2B: $\chi^2(3) = 0.368$, $p = 0.868$ on day 1 and $\chi^2(3) = 4.782$, $p = 0.104$ on day 3, pNR2B: $\chi^2(3) = 0.644$, $p = 0.818$ on day 1 and $\chi^2(3) = 0.644$, $p = 0.818$ on day 3). However, the levels of total NR2B and pNR2B were significantly different among the normal, noise + vehicle, and noise + 2 ml LiCl groups on days 7 after the noise exposure and/or the treatment (NR2B 004 and pNR2B: $\chi^2(3) = 7.448$, $p = 0.004$). The levels of total NR2B and pNR2B were increased 7 days after the noise exposure, and that noise-induced increase in total NR2B and pNR2B expression were suppressed by round-window delivery of LiCl. The post-hoc analysis with Dunn's multiple comparison test showed that the noise + 2 mM LiCl group had significantly lower NR2B and pNR2B expression than the noise + vehicle group 7 days after the treatment (NR2B and pNR2B: $p = 0.0019$).

## Discussion

Our study has demonstrated that round-window delivery of high-dose lithium after a noise exposure rescued the noise-induced loss of presynaptic ribbons (Fig 4) and restored the supra-threshold amplitudes for ABR wave 1 at frequencies above the noise band (Fig 3). In addition,

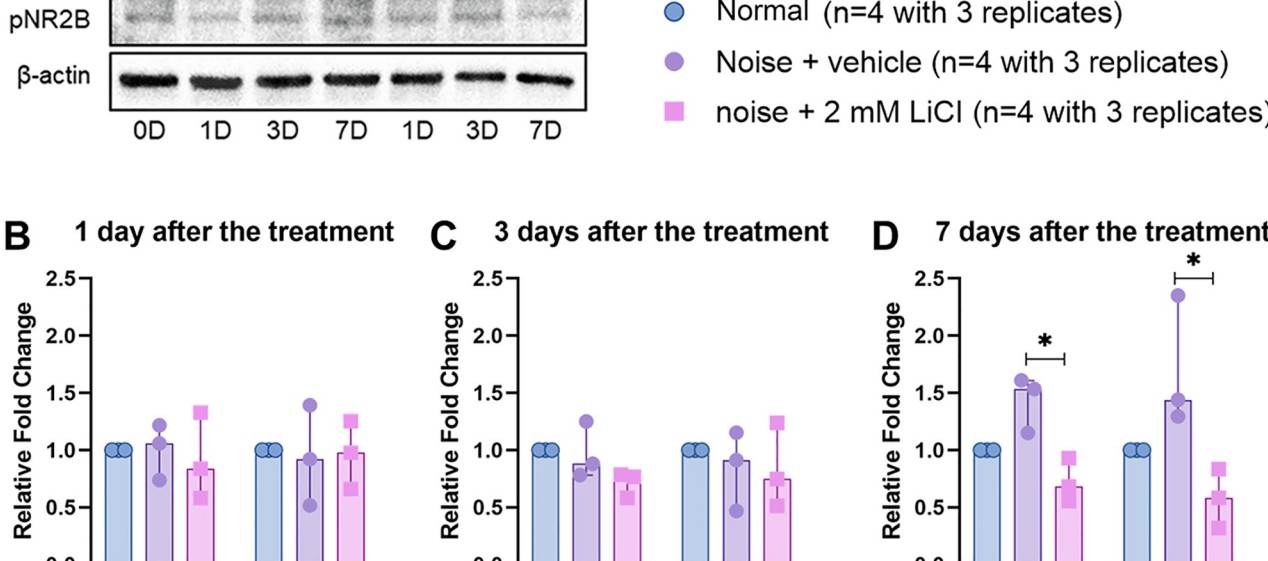

**Fig 5. A** Western blotting analysis of total NR2B, phospho-NR2B (pNR2B), β-actin protein expression levels in the cochlea. β-actin is the loading control. **B-D** Separated scatter graph representing the fold changes for the total NR2B and pNR2B band. The fold changes for each protein normalized to *β*-actin and then compared with normal group, are indicated. Bar graph depicting medians and 95% confidence interval around the median. Statistical analysis of the fold changes was conducted among the normal, noise + vehicle, and noise + 2 ml LiCl groups. Statistical significance is indicated by asterisks: * $p < 0.05$ based on a post-hoc analysis with Dunnett's multiple comparisons test.

lithium delivery led to early recovery of the ABR threshold shifts after noise-induced damage (Fig 2). There is a robust literature on the delivery of potential therapeutic agents to prevent or regenerate noise-induced synaptic loss in the cochlea [7, 23–26]. However, few studies have successfully demonstrated regeneration of cochlear synapses when a therapeutic agent was administered after noise exposure, such as our study [27].

There is strong evidence that an excess release of the excitatory neurotransmitter glutamate from the IHC synapses in response to acoustic overstimulation may cause excitotoxicity with a loss of synaptic connections to the SGN [5, 6, 9–12]. Glutamate excitotoxicity can allow calcium influx which in turn can trigger cell death pathways eventually leading to auditory hair cell death [28]. Although the pharmacological mechanisms of lithium are not completely understood, lithium has been reported to have neuroprotective effects against the glutamate-induced excitotoxicity [16–18, 29]. The neuroprotective effect of lithium has been associated with the suppression of NR2B phosphorylation of the NMDA glutamate receptor, inhibition of NMDA receptor–mediated calcium influx, upregulation of anti-apoptotic Bcl-2, downregulation of pro-apoptotic p53 and Bax, and activation of the survival signaling pathway [18, 29, 30]. Our western blotting results indicate that noise exposure induced increases in the expression of the NR2B subunit and its phosphorylation at Ser1303 after 7 days, but round-window delivery of high-dose lithium suppressed these NR2B levels (Fig 4).

In the mammalian inner ear, SGNs express a variety of glutamate receptor types, including NMDA receptors and α-amino-3-hydroxy-5-methylisoxazole-4-propionic acid (AMPA)

receptors [31]. AMPA receptors mediate the large fast excitatory currents necessary for synaptic transmission and thereby maintain the temporal fidelity essential for hearing [32, 33], whereas the NMDA receptors are not involved in fast excitatory neurotransmission because they are blocked at resting potential by $Mg^{2+}$. NMDA receptors are instead activated by excessive neurotransmitter released from IHCs that becomes excitotoxic to the primary auditory neuron [34–36]. NMDA receptors are not directly involved in glutamatergic transmission at IHC–SGN synapses [33], but they are thought to contribute to the regulation of AMPA receptor expression following acute insult [37, 38]. In mouse auditory neurons, the expression of surface AMPA receptors decreased in the presence of an NMDA receptor agonist, and that decrease was blocked by the application of an NMDA receptor antagonist [37]. Under physiological conditions, post-synaptic AMPA receptors are paired with pre-synaptic ribbons. However, studies of noise-induced hearing loss have shown that reorganization of synaptic ribbon locations causes orphan AMPA receptors that lack opposed pre-synaptic ribbons and the loss of synaptic ribbons [39, 40]. In normal mature mouse cochleae, most NMDA receptors are distributed on the modiolar side close to the nuclear region of the IHCs, whereas most synaptic ribbons and AMPA receptors are on neural terminals closer to the basal poles of the IHCs. After gentamicin exposure, AMPA receptors and NMDA receptors are relocated to nerve fiber terminals around the IHCs. The NMDA receptors move downward toward the basal poles of the IHCs, and the AMPA receptors move upward toward the bundle poles of the IHCs [41]. Those results suggest that the post-synaptic rearrangement of AMPA receptors, modulated by NMDA receptors, might affect the number and location of pre-synaptic ribbons. Therefore, inhibiting NMDA receptors might prevent noise-induced synaptic loss by preventing the relocation of AMPA and NMDA receptors on the dendrites of the SGNs and thus maintain the integrity of the ribbon synapses, preserving hearing function.

Although the function of NMDA receptors in noise-induced cochlear synaptopathy is unclear, several studies have suggested that NMDA receptors play a role in the excitotoxicity caused by noise trauma, and its antagonists have shown potential as treatments for noise-induced cochlear synaptopathy [34]. Calcium influx through NMDA receptor at excitatory synapse causes activation of post-synaptic calcium/calmodulin-dependent protein kinase type II (CaMKII) and CaMKII undergoes phosphorylation at Ser1301 by binding to the NR2B subunit of NMDA receptor [42]. Thus, lithium treatment likely has neuroprotective effects against noise-induced excitotoxicity by the suppression of inhibition of NMDA receptor–mediated calcium influx.

Our study has some limitations. First, round-window delivery of LiCl could yield inconsistent results. It was difficult to deliver a viscous gel into the small bulla opening required to minimize the surgically induced threshold shift. That constraint likely decreased the effective dose of lithium chloride delivered to the round-window. Nevertheless, lithium was administered locally in this study because it has a narrow therapeutic window and adverse effects on the kidneys, thyroid gland, and parathyroid glands, which necessitates close monitoring of its plasma concentration and the functioning of those organs. Second, we initially intended to make a rat model of noise-induced synaptic damage without causing permanent elevation of the hearing threshold. Prior work on this type noise-induced synaptic damage caused no significant permanent threshold elevation [43], but our model did cause permanent threshold elevations of ~15 dB SPL at 16 and 32 kHz without a significant loss of hair cells (Fig 2). Because noise-induced threshold shifts are sensitive to overall exposure energy, it seems necessary to adjust the exposure level or duration. Third, the pharmacological mechanisms by which lithium regenerates IHC synapses are still not fully understood. Lithium causes a wide range of intracellular responses that affect the neurotrophic response, autophagy, oxidative stress, inflammation, and mitochondrial function [44–49]. The blockade of NMDA receptor

activity by lithium could occur through any of several mechanisms to prevent the loss of IHC synapses, so further study is needed. Fourthly, many key questions remain for clinical application. There is likely a significant clinical population with loss of cochlea synapse that could be addressed by a treatment analogous to that used here in the noise-exposed rat. Although the local delivery of lithium to the round window was effective in regenerating cochlear synapses 24 hours after the noise exposure, a key question would be how long this time window could be extended for effective synaptogenesis in a clinical setting.

## Conclusion

The present experiments showed that round window delivery of lithium can be effective local access route to the inner ear, and regenerates cochlear synapses even when administered 24 hours post-exposure of noise.

## Supporting information

**S1 Fig. Western blot analysis of total NR2B, phospho-NR2B (pNR2B), β-actin protein expression levels in the cochlea.** A-C Original uncropped and unadjusted images representing for the β-actin (A), total NR2B (B), and phopho-NR2B (C). d = day, R = replicate. (DOCX)

## Author Contributions

**Conceptualization:** Ji Eun Choi, Jae-Hun Lee, Min Young Lee.

**Data curation:** Nathaniel T. Carpena.

**Formal analysis:** Ji Eun Choi.

**Funding acquisition:** Ji Eun Choi.

**Investigation:** Ji Eun Choi, Nathaniel T. Carpena, Jae-Hun Lee.

**Methodology:** Ji Eun Choi, Nathaniel T. Carpena, Jae-Hun Lee, So-Young Chang.

**Project administration:** Ji Eun Choi.

**Supervision:** So-Young Chang, Min Young Lee, Jae Yun Jung, Won-Ho Chung.

**Visualization:** Nathaniel T. Carpena.

**Writing – original draft:** Ji Eun Choi.

**Writing – review & editing:** Jae Yun Jung, Won-Ho Chung.

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
