## [Decision Letter · Decision Letter 0]

31 Oct 2022

PONE-D-22-25555Round-window delivery of lithium chloride protects cochlear synapses from noise-induced excitotoxic trauma by inhibiting NMDA receptorPLOS ONE

Dear Dr. Jung,

Thank you for submitting your manuscript to PLOS ONE. The manuscript has been reviewed, reviews attached, and after careful consideration, we feel that it has merit but does not fully meet PLOS ONE’s publication criteria as it currently stands. Therefore, we invite you to submit a revised version of the manuscript that addresses the points raised during the review process.

The manuscript is of interest, but the reviewers have raised a number of concerns that need to be addressed.  Some of these listed are relatively minor and I recommend you address them but there are a number of more significant concerns about the statistical analyses and the lack of a control group, which both need to be considered and addressed in any revision before we can consider accepting the paper for publication.

We look forward to receiving your revised manuscript.

Kind regards,

Peter Thorne, PhD

Academic Editor

PLOS ONE

Journal Requirements:

"This work was supported by a National Research Foundation of Korea (NRF) grant funded by the Korean government (MSIT) (no. NRF-2020R1C1C1009849) and by the Basic Science Research Program through the NRF founded by the Ministry of Education (NRF-2020R1A6A1A03043283). The funders had no role in study design, data collection and analysis, decision to publish, or preparation of the manuscript."

"This work was supported by a National Research Foundation of Korea (NRF) grant funded by the Korean government (MSIT) (no. NRF-2020R1C1C1009849) and by the Basic Science Research Program through the NRF founded by the Ministry of Education (NRF-2020R1A6A1A03043283). The funders had no role in study design, data collection and analysis, decision to publish, or preparation of the manuscript."

5. To comply with PLOS ONE submissions requirements, in your Methods section, please provide additional information regarding the experiments involving animals and ensure you have included details on (1) methods of sacrifice, (2) methods of anesthesia and/or analgesia, and (3) efforts to alleviate suffering.

6. As part of your revision, please complete and submit a copy of the Full ARRIVE 2.0 Guidelines checklist, a document that aims to improve experimental reporting and reproducibility of animal studies for purposes of post-publication data analysis and reproducibility: https://arriveguidelines.org/sites/arrive/files/documents/Author%20Checklist%20-%20Full.pdf (PDF). Please include your completed checklist as a Supporting Information file. Note that if your paper is accepted for publication, this checklist will be published as part of your article.

7. Please upload a copy of Supporting Information Figure S1 to which you refer to in your text on page 16.

Reviewers' comments:

Reviewer's Responses to Questions

**Comments to the Author**

1. Is the manuscript technically sound, and do the data support the conclusions?

Reviewer #1: Yes

Reviewer #2: Partly

2. Has the statistical analysis been performed appropriately and rigorously? 

Reviewer #1: Yes

Reviewer #2: No

3. Have the authors made all data underlying the findings in their manuscript fully available?

Reviewer #1: Yes

Reviewer #2: Yes

4. Is the manuscript presented in an intelligible fashion and written in standard English?

Reviewer #1: Yes

Reviewer #2: No

5. Review Comments to the Author

Reviewer #1: This is a well written paper on the application of a LiCl treatment for the protection of the hair cell-auditory neuron synapse following a noise exposure in rats. The results would be of interest to researchers and clinicians in the field of hearing research.

I have a few specific comments that should be addressed:

The abstract mentions synaptic regeneration, but later talks about reducing synaptic loss. In this paper, it is more likely to be the latter that is occurring, since the treatment is given 24 hours after noise exposure. Please check the manuscript to make sure there are no other inconsistencies with respect to protection/regeneration.

The introduction was appropriately reference and introduced all relevant topics.

In this sentence in the methods: “The control group was not exposed to noise or surgery to access normal cochlear synaptic counts.” Do you mean assess here?

It’s not clear from the methods whether the lithium is injected through the RWM or into the niche. Please amend the text to clarify.

Figure 1 is a nice summary of the experimental protocol. Thanks for including this.

In this sentence in the results: “As quantitatively shown in Fig. 3C and D, the mean synaptic count per IHC was reduced”, do you mean Fig 4C and D?

And here as well: “Synaptic rescue can be seen in the noise + 2 mM LiCl group both one week (Fig. 3A) and two weeks (Fig. 3B) after the treatment.” I think you mean Fig 4A and B

In figure 4, there are some differences in the brightness of the green stain between images, particularly in Fig 4B noise + vehicle which seems dimmer than the ones on either side of it. It’s hard to see the synaptic puncta. Also in Fig 4A noise + vehicle and noise + 2 mM LiCl. Could these images be adjusted to be of similar intensity?

In figure 5 there are 2 bands in the NR2B image. Are they both relevant to the protein? Do you expect to see a reduction in NR2B compared to control? In the results section you only compare to noise-induced increase in total NR2Band pNR2B, but what would you find if you compared to control?

The first sentence in the discussion seems a little off topic, mentioning a therapeutic window in which to regenerate “peripheral axons”, whereas this study is looking at synapses.

In the discussion – Could you please include discussion on the timing of the experimental protocol, and the clinical applicability of delivering a therapeutic within 1 day of a noise exposure, particularly a noise exposure that only causes synaptopathy, as opposed to one that causes a significant threshold shift, and particularly where the treatment requires deliver of the agent in a clinical setting (as opposed to taking an oral medication for example). If synaptopathy in the human population occurs because of moderate noise exposure over a long period of time, how applicable is the animal model used in this study? How would the patient identify that a damaging noise exposure has occurred?

Depending on the requirements of the journal, a conclusion would be nice to have.

Reviewer #2: This manuscript investigated the effects of round-window delivery of lithium on noise-induced cochlear synaptopathy and hearing loss in a rat model. They also investigated the changes in NMDA receptor expression as potential underlying mechanisms. The experimental design is straightforward, although including a control + vehicle group would add more confidence for some of the conclusions (see specific comments below). The methods were appropriate and well described except the statistical analysis (see specific comments below). The results section could be improved with a more appropriate statistical analysis.

Since there is no page numbers and line numbers for the downloaded manuscript, the page numbers mentioned in the comments below is referred to the page number of the entire PDF file , which does not corresponding to the actual page numbers of the manuscript.

1. Title: The animal species needs to be included in the title

2. Page 9, Abstract:

a. Please provide the dose (or dose range) of lithium chloride and clarify whether it was a single treatment or multiple treatment

b. Throughout the whole abstract, there is no mentioning of what animals was used.

3. Page 10: “it could contribute to difficulties understanding …” should be “…difficulties in understanding…”.

4. Page 14, first line: Will the ABR testing at 3 days after treatment not only assess the “effects of the surgical manipulations”, but also the treatment effects?

5. Page 14, line 6: n = 2 for groups 1 and 2?

6. Page 15, line 1: What does “imaged at frequencies of 8, 12, 16 and 32 kHz” mean? Please clarify how the frequency location was determined.

7. Page 15, Western blot analysis: The statement “To assess NR2B activity” is not appropriate, as Western blot can only look at the protein expression of the receptor, but not the activity of the receptor.

8. Page 16, line 15: “followed by autoradiography”. “Autoradiography” is referred to the photochemical technique used to record the spatial distribution of radiolabeled compounds within a specimen or an object, which shouldn’t be used to describe the imaging analysis procedure for Western blot using ChemiDoc.

9. Pages 16 – 17: “To compare the ABR thresholds, synapse counts, and western blot result between experimental groups.” This is not a complete sentence.

10. Page 17: The sentence “parametric data were analyzed using the repeated measures one-way analysis of variance (ANOVA) was used with the Greenhouse-Geisser correction” is not grammatically correct.

11. Statistical analysis:

a. For the ABR data, you have 5 groups (control, noise-only, noise + vehicle, noise + 1mM and noise + 2mM), 4 time pints (baseline, 3 days, 1 week and 2 weeks) and 4 frequencies (8, 12, 16 and 32). A 3-way ANOVA (group x time x frequency) with repeated measures or a linear mixed model analysis should be used. If you want to analyse each frequency separately, you should use a two-way ANOVA (group x time) with repeated measures or a linear mixed model analysis. An one-way ANOVA is not appropriate for this data. In addition, why the baseline data was not included in the analysis?

b. For the wave I amplitude data (i.e., figure 3), no statistical method was mentioned. Assume that a one-way ANOVA was used, but looking at the data, at least a two-way ANOVA (time x stimulus level) with repeated measures should be used for each group instead, but a 3- way analysis would be even better.

12. Page 17, Results: The statement of “a temporary ~15 dB SPL threshold shift and a permanent ~15 dB SPL threshold shift” is not appropriate, as “temporary” means lasting for only a limited period of time; not permanent, but the threshold shift was “permanent” in your case. So, please rewrite this sentence.

13. Figure 2: The baseline ABR for all the groups should be included in order to see changes before and after noise exposure/drug treatment.

14. Page 17, Results: “The surgery for the round-window delivery of vehicle caused temporary increases in the 8 kHz and 12 kHz thresholds for only one week after the surgery, as observed by comparing the noise-only and noise + vehicle groups”. Without a control + vehicle group, the conclusion that “the surgery for the round-window delivery of vehicle caused temporary increases in the 8 kHz and 12 kHz thresholds” cannot be made. As shown in Fig 2, there seems no difference in the threshold between 3 days and 1 week for the noise + vehicle group (there seems slight improvement).

15. Page 18, Results: “All experimental groups had statistically significant differences before and after noise exposure as determined by one-way ANOVA with the Greenhouse-Geisser correction for all levels from 65 -80 dB SPL”. This statement is not backed up by the one-way ANOVA analysis, as you were talking about two factors here, i.e., time and levels. It is not clear how the one-way ANOVA could be run to tell you the differences for the two factors. As commented on the statistical analysis, a two-way ANOVA or 3-way ANOVA/linear mixed model should be used.

16. Figure 3 legend: what does the grey area mean? The exposed intensity? Please clearify.

17. Page 19, Results: For the synapses data, it is also better to do a 2-way ANOVA or linear mixed model analysis for similar reasons mentioned above. In addition, n = 2 seems a very small sample size for valid statistical analysis.

18. Figure 5: please add sample size for each group. From the western blot image, there seems only one sample in the control group and 2 samples in the other two groups. Please clarify.

19. The Supplementary figure seems not necessary.

6. PLOS authors have the option to publish the peer review history of their article (what does this mean?). If published, this will include your full peer review and any attached files.

Reviewer #1: **Yes: **Rachael T Richardson

Reviewer #2: **Yes: **Yiwen Zheng

---

## [Author Response · Author response to Decision Letter 0]

30 Nov 2022

Review Comments to the Author:

Reviewer #1: This is a well written paper on the application of a LiCl treatment for the protection of the hair cell-auditory neuron synapse following a noise exposure in rats. The results would be of interest to researchers and clinicians in the field of hearing research.

I have a few specific comments that should be addressed:

#1. The abstract mentions synaptic regeneration, but later talks about reducing synaptic loss. In this paper, it is more likely to be the latter that is occurring, since the treatment is given 24 hours after noise exposure. Please check the manuscript to make sure there are no other inconsistencies with respect to protection/regeneration.

Author’s Response: Thank you for your comment. We made sure to use the term “regeneration” throughout the entire manuscript. 

#2. The introduction was appropriately reference and introduced all relevant topics.

Author’s Response: Thank you for positive response.

#3. In this sentence in the methods: “The control group was not exposed to noise or surgery to access normal cochlear synaptic counts.” Do you mean assess here?

Author’s Response: It means negative control. Some modifications have been made to avoid confusion to readers as follows: The control group was not exposed to noise or surgery (Group 1: control group, n=8) (Line 83-84). 

#4. It’s not clear from the methods whether the lithium is injected through the RWM or into the niche. Please amend the text to clarify.

Author’s Response: Thank you for your valid point. The sentence has been changes as follows: A 16G needle was positioned within the round-window niche, and 50 µl of the poloxamer solution was injected using a 1 mL syringe over the round-window membrane (Line 120-121).

#5. Figure 1 is a nice summary of the experimental protocol. Thanks for including this.

Author’s Response: Thank you for positive response.

#6. In this sentence in the results: “As quantitatively shown in Fig. 3C and D, the mean synaptic count per IHC was reduced”, do you mean Fig 4C and D?

And here as well: “Synaptic rescue can be seen in the noise + 2 mM LiCl group both one week (Fig. 3A) and two weeks (Fig. 3B) after the treatment.” I think you mean Fig 4A and B

Author’s Response: Thank you for your careful observation. We changed the number. 

#7. In figure 4, there are some differences in the brightness of the green stain between images, particularly in Fig 4B noise + vehicle which seems dimmer than the ones on either side of it. It’s hard to see the synaptic puncta. Also in Fig 4A noise + vehicle and noise + 2 mM LiCl. Could these images be adjusted to be of similar intensity?

Author’s Response: Thank you for your comment. We increased the brightness of Noise + vehicle and noise + 2 mM LiCl in Fig. 4A and Noise + vehicle in Fig. 4B.

#8. In figure 5 there are 2 bands in the NR2B image. Are they both relevant to the protein? Do you expect to see a reduction in NR2B compared to control? In the results section you only compare to noise-induced increase in total NR2Band pNR2B, but what would you find if you compared to control?

Author’s Response: Thank you for your comment. The other papers also showed 2 bands in the NR2B image (see reference Neurosignals . 2013;21(1-2):42-54, PMID: 22377595). We expected that the levels of total NR2B and pNR2B would be increased after the noise exposure, and that noise-induced increase in total NR2B and pNR2B expression, and that noise-induced increase in total NR2B and pNR2B expression would be suppressed by lithium treatment. The post-hoc analysis with Dunn’s multiple comparison test showed that the noise + 2 mM LiCl group had significantly lower NR2B and pNR2B expression than the noise + vehicle group 7 days after the treatment (NR2B and pNR2B: p = 0.0019) (Line 292-295).

#9. The first sentence in the discussion seems a little off topic, mentioning a therapeutic window in which to regenerate “peripheral axons”, whereas this study is looking at synapses.

Author’s Response: Thank you for your valid point. We deleted the first sentence. 

#10. In the discussion – Could you please include discussion on the timing of the experimental protocol, and the clinical applicability of delivering a therapeutic within 1 day of a noise exposure, particularly a noise exposure that only causes synaptopathy, as opposed to one that causes a significant threshold shift, and particularly where the treatment requires deliver of the agent in a clinical setting (as opposed to taking an oral medication for example). If synaptopathy in the human population occurs because of moderate noise exposure over a long period of time, how applicable is the animal model used in this study? How would the patient identify that a damaging noise exposure has occurred?

Author’s Response: Thank you for your comment. We included discussion on the timing of the treatment as follows: There is a robust literature on potential therapeutic agent to prevent or regenerate noise-induced synaptic loss in the cochlea [7, 23-26]. However, few studies have successfully demonstrated regeneration of cochlear synapses when a therapeutic agent was administered after noise exposure, such as our study [27] (Line 303-306). Also, we included discussion on the application to human hearing impairment at the end of the discussion as follows: Fourthly, many key questions remain for clinical application. There is likely a significant clinical population with loss of cochlea synapse that could be addressed by a treatment analogous that used here in noise-exposed rat. Although the success of local lithium delivery in regenerating cochlear synapses, most important is how long the therapeutic window can be elicited synaptogenesis after the noise exposure. (Line 371-375).

#11. Depending on the requirements of the journal, a conclusion would be nice to have.

Author’s Response: Thank you for your comment. We added the conclusion as follows: The present experiments showed that round window delivery of lithium can be effective local access route to the inner ear, and regenerates cochlear synapses even when administered 24 hours post-exposure of noise.

Reviewer #2: This manuscript investigated the effects of round-window delivery of lithium on noise-induced cochlear synaptopathy and hearing loss in a rat model. They also investigated the changes in NMDA receptor expression as potential underlying mechanisms. The experimental design is straightforward, although including a control + vehicle group would add more confidence for some of the conclusions (see specific comments below). The methods were appropriate and well described except the statistical analysis (see specific comments below). The results section could be improved with a more appropriate statistical analysis.

Since there is no page numbers and line numbers for the downloaded manuscript, the page numbers mentioned in the comments below is referred to the page number of the entire PDF file , which does not corresponding to the actual page numbers of the manuscript.

#1. Title: The animal species needs to be included in the title

Author’s Response: Thank you for your comment. The title has been changed as follows: Round-window delivery of lithium chloride protects rat cochlear synapses from noise-induced excitotoxic trauma by inhibiting NMDA receptor

#2. Page 9, Abstract:

a. Please provide the dose (or dose range) of lithium chloride and clarify whether it was a single treatment or multiple treatment

b. Throughout the whole abstract, there is no mentioning of what animals was used.

Author’s Response: Thank you for your valuable comment. We provided the dose and number of treatments of lithium chloride in the abstract as follows: We locally delivered a single treatment of poloxamer 407 (vehicle) containing lithium chloride (either 1 mM or 2 mM) to the round-window niche 24 hours after noise exposure (Line 28-29). Also, we mentioned what animal was used as follows: Our rat animal model of noise-induced cochlear synaptopathy caused about 50% loss of synapses in the cochlear basal region without damaging hair cells (Line 26-27).

#3. Page 10: “it could contribute to difficulties understanding …” should be “…difficulties in understanding…”.

Author’s Response: Thank you for your careful observation. We added the “in” in the line 49. 

#4. Page 14, first line: Will the ABR testing at 3 days after treatment not only assess the “effects of the surgical manipulations”, but also the treatment effects?

Author’s Response: Thank you for this valid point. That sentence has been deleted because it can confuse readers. The ABR test after 3 days of treatment assessed the “effects of the surgical manipulations” as well as the “treatment effects”. (Line 126-138)

#5. Page 14, line 6: n = 2 for groups 1 and 2?

Author’s Response: Group 1 and 2 were assigned two rats each. We revised the wording as follows: n=2 for group 1 and 2, n=5 for group 3-5 at 1 week and 2 weeks post-treatment in the Line 141-142.

#6. Page 15, line 1: What does “imaged at frequencies of 8, 12, 16 and 32 kHz” mean? Please clarify how the frequency location was determined.

Author’s Response: Thank you for your comment. The location of the basilar membrane at the relevant frequency was determined by computing a cochlear frequency map (Muller, 1991 reference #22). Since the end of the sentence describes how the frequency location was determined, the first sentence has been modified as follows. Images were obtained using a confocal microscope (Flow-View 3000, Olympus, Japan) with a glycerol-immersion 40X objective and 2X digital zoom.

#7. Page 15, Western blot analysis: The statement “To assess NR2B activity” is not appropriate, as Western blot can only look at the protein expression of the receptor, but not the activity of the receptor.

Author’s Response: Thank you for your comment. The wording has been changed as follows: To perform the western blot (Line 175).

#8. Page 16, line 15: “followed by autoradiography”. “Autoradiography” is referred to the photochemical technique used to record the spatial distribution of radiolabeled compounds within a specimen or an object, which shouldn’t be used to describe the imaging analysis procedure for Western blot using ChemiDoc.

Author’s Response: Thank you for your valid point. The sentence has been changed as follows: After being thoroughly washed with PBST, the membranes were visualized by enhancing chemiluminescence substrate (Pierce, Rockford, IL) for 2 min, followed by chemiluminescence detection on ChemiDoc XRS+ System (Bio-Rad Laboratories, Hercules, CA, USA) (Line 195-197).

#9. Pages 16 – 17: “To compare the ABR thresholds, synapse counts, and western blot result between experimental groups.” This is not a complete sentence.

Author’s Response: The reviewer’s comments are appropriate. The sentence has been changed as follows: The ABR thresholds, ABR wave 1 amplitude, synapse counts, and western blot result were compared between experimental groups (Group 3-5) (Line 206-207).

#10. Page 17: The sentence “parametric data were analyzed using the repeated measures one-way analysis of variance (ANOVA) was used with the Greenhouse-Geisser correction” is not grammatically correct.

Author’s Response: The reviewer’s comments are appropriate. The sentence has been changed as follows: For multiple groups, nonparametric data were statistically analyzed using the Kruskal-Wallis H test and the repeated measures one-way analysis of variance (ANOVA) with the Greenhouse-Geisser correction.

#11. Statistical analysis:

a. For the ABR data, you have 5 groups (control, noise-only, noise + vehicle, noise + 1mM and noise + 2mM), 4 time pints (baseline, 3 days, 1 week and 2 weeks) and 4 frequencies (8, 12, 16 and 32). A 3-way ANOVA (group x time x frequency) with repeated measures or a linear mixed model analysis should be used. If you want to analyse each frequency separately, you should use a two-way ANOVA (group x time) with repeated measures or a linear mixed model analysis. An one-way ANOVA is not appropriate for this data. In addition, why the baseline data was not included in the analysis?

Author’s Response: We fully understand the reviewer’s concerns. As mentioned in the statistical analysis, the ABR thresholds was compared between experimental groups (Group 3-5). Thus, we have 3 groups (noise + vehicle, noise + 1mM and noise + 2mM), 4 time points (baseline, 3 days, 1 week and 2 weeks) and 4 frequencies (8, 12, 16 and 32). For showing the ABR threshold result, usually, the Y-axis represents ABR thresholds, and the X-axis represents frequency of stimuli presented. To show the ABR thresholds between the experimental groups, audiograms for ABR threshold were separated by time point. All rats have normal hearing before surgery (baseline), so that the ABR threshold at baseline was omitted. Based on the figures, we analyzed the ABR threshold each frequency and each time point separately using Kruskal-Wallis H test. 

b. For the wave I amplitude data (i.e., figure 3), no statistical method was mentioned. Assume that a one-way ANOVA was used, but looking at the data, at least a two-way ANOVA (time x stimulus level) with repeated measures should be used for each group instead, but a 3- way analysis would be even better.

Author’s Response: We fully understand the reviewer’s concerns. We mentioned the statistical method in line 243-244 as follows: All experimental groups had statistically significant differences before and after noise exposure as determined by one-way ANOVA with the Greenhouse-Geisser correction for all levels from 65–80 dB SPL. We want to see the changes of ABR wave 1 amplitude from 65-80dB SPL in each group, which were measured multiple times. Here, the dependent variable is one (time). If we used two-way ANOVA (4 times x 4 stimulus level), we could get the results of multiple comparison test depending on each stimulus level. As reviewer’s comment, we re-analyzed using two-way ANOVA. 

#12. Page 17, Results: The statement of “a temporary ~15 dB SPL threshold shift and a permanent ~15 dB SPL threshold shift” is not appropriate, as “temporary” means lasting for only a limited period of time; not permanent, but the threshold shift was “permanent” in your case. So, please rewrite this sentence.

Author’s Response: We fully understand the reviewer’s concerns. As shown in figure 2, noise exposure caused 30 dB SPL threshold shift until one week after the treatment. Two weeks after the treatment, there was improvement of 15 dB SPL threshold. Thus, finally, noise exposure caused a permanent ~15 dB SPL threshold shift. The sentence has been changed as follows: As observed by comparing the control and noise-only groups, noise exposure caused a threshold shift of ~30 dB SPL at frequencies above the noise band, when measured 1 week after the treatment, (Fig. 2). Two week later, ABR thresholds recovered up to 15 dB SPL. Finally, noise exposure caused a permanent ~15 dB SPL threshold shift at frequencies above the noise band (Lone 216-220)

#13. Figure 2: The baseline ABR for all the groups should be included in order to see changes before and after noise exposure/drug treatment.

Author’s Response: The reviewer’s comments are appropriate. We added the baseline ABR threshold in the figure 2-A.

#14. Page 17, Results: “The surgery for the round-window delivery of vehicle caused temporary increases in the 8 kHz and 12 kHz thresholds for only one week after the surgery, as observed by comparing the noise-only and noise + vehicle groups”. Without a control + vehicle group, the conclusion that “the surgery for the round-window delivery of vehicle caused temporary increases in the 8 kHz and 12 kHz thresholds” cannot be made. As shown in Fig 2, there seems no difference in the threshold between 3 days and 1 week for the noise + vehicle group (there seems slight improvement).

Author’s Response: Thank you for your comment. The difference between the noise-only and noise + vehicle groups is whether the vehicle, a thermos-reversible gel, was surgically placed into the middle ear cavity. Therefore, it can be assumed that the difference in ABR threshold between the two groups is due to the presence of the thermos-reversible gel in the middle ear cavity. Conductive hearing loss could be caused by thermos-reversible gel by 1 week after the treatment. The sentence has been changed as follows: Compared to the noise-only group, the noise + vehicle group had worse ABR threshold at 8 kHz and 12 kHz at 1 week after the treatment, but ABR threshold in the noise + vehicle group had returned to that in the noise-only group (Line 223-224).

#15. Page 18, Results: “All experimental groups had statistically significant differences before and after noise exposure as determined by one-way ANOVA with the Greenhouse-Geisser correction for all levels from 65 -80 dB SPL”. This statement is not backed up by the one-way ANOVA analysis, as you were talking about two factors here, i.e., time and levels. It is not clear how the one-way ANOVA could be run to tell you the differences for the two factors. As commented on the statistical analysis, a two-way ANOVA or 3-way ANOVA/linear mixed model should be used.

Author’s Response: Thank you for your comment. As response to comment #11-b, we used a two-way ANOVA (Line 244-257).

#16. Figure 3 legend: what does the grey area mean? The exposed intensity? Please clearify.

Author’s Response: Thank you for your comment. The gray shaded area indicates suprathreshold stimuli (all levels from 65–80 dB SPL). We added the sentence in the figure 3 legend.

#17. Page 19, Results: For the synapses data, it is also better to do a 2-way ANOVA or linear mixed model analysis for similar reasons mentioned above. In addition, n = 2 seems a very small sample size for valid statistical analysis.

Author’s Response: We fully understand the reviewer’s concerns. As mentioned in the statistical analysis, the synaptic count was compared between experimental groups (Group 3-5). Also, synaptic count is not repeated measured data. We used a Kruskal-Wallis H test. 

#18. Figure 5: please add sample size for each group. From the western blot image, there seems only one sample in the control group and 2 samples in the other two groups. Please clarify.

Author’s Response: Thank you for your comment. As mentioned in the method, four cochlear tissues per group were pooled with 3 replicates. We changed the figure 5. 

#19. The Supplementary figure seems not necessary.

Author’s Response: Thank you for your comment. The supplementary figure was deleted.

---

## [Decision Letter · Decision Letter 1]

24 Feb 2023

PONE-D-22-25555R1Round-window delivery of lithium chloride regenerates rat cochlear synapses from noise-induced excitotoxic trauma by inhibiting NMDA receptorPLOS ONE

Dear Dr. Jung,

Thank you for submitting your manuscript to PLOS ONE. After careful consideration, we feel that it has merit but does not fully meet PLOS ONE’s publication criteria as it currently stands. Therefore, we invite you to submit a revised version of the manuscript that addresses the points raised during the review process.

ACADEMIC EDITOR: Please address the minor changes recommended by both reviewers, including the comparison between control and experimental groups.

We look forward to receiving your revised manuscript.

Kind regards,

Alan G. Cheng, M.D.

Academic Editor

PLOS ONE

Journal Requirements:

Reviewers' comments:

Reviewer's Responses to Questions

**Comments to the Author**

1. If the authors have adequately addressed your comments raised in a previous round of review and you feel that this manuscript is now acceptable for publication, you may indicate that here to bypass the “Comments to the Author” section, enter your conflict of interest statement in the “Confidential to Editor” section, and submit your "Accept" recommendation.

Reviewer #1: All comments have been addressed

Reviewer #2: (No Response)

2. Is the manuscript technically sound, and do the data support the conclusions?

Reviewer #1: Yes

Reviewer #2: Partly

3. Has the statistical analysis been performed appropriately and rigorously? 

Reviewer #1: Yes

Reviewer #2: No

4. Have the authors made all data underlying the findings in their manuscript fully available?

Reviewer #1: Yes

Reviewer #2: Yes

5. Is the manuscript presented in an intelligible fashion and written in standard English?

Reviewer #1: No

Reviewer #2: Yes

6. Review Comments to the Author

Reviewer #1: Thank you for the changes you made to the manuscript.

I have a couple of minor revisions that I think need to be addressed prior to publication.

Suggested change to the title from "Round-window delivery of lithium chloride regenerates rat cochlear synapses from noise-induced excitotoxic trauma by inhibiting NMDA receptor" to one of the following:

"Round-window delivery of lithium chloride regenerates cochlear synapses damaged by noise-induced excitotoxic trauma via inhibition of the NMDA receptor in the rat"

OR

"Regeneration of cochlear synapses in the rat following noise-induced excitotoxic trauma by round-window delivery of lithium chloride to inhibit the NMDA receptor" or something similar, because I don't think "regenerates rat cochlear synapses from noise-induced excitotoxic trauma" makes sense.

Line 208: "the two-way repeated measures one-way analysis of variance" Was it one way or two way?

Line 224 "in the noise + vehicle group had returned to that in the noise-only group." change to ...of the noise-only group.

Line 302 "There is a robust literature on potential therapeutic agent to prevent or regenerate noise-induced synaptic loss in the cochlea [7, 23-26]. " Change to "There is a robust literature on the delivery of potential therapeutic agents to prevent or regenerate noise-induced synaptic loss in the cochlea [7, 23-26].

Suggested change to Line 371 ""Fourthly, many key questions remain for clinical application. There is likely a significant clinical population with loss of cochlea synapse that could be addressed by a treatment analogous to that used here in the noise-exposed rat. Although the local delivery of lithium to the round window was effective in regenerating cochlear synapses 24 hours after the noise exposure, a key question would be how long this time window could be extended for effective synaptogenesis in a clinical setting."

Reviewer #2: The authors have addressed most of my comments to my satisfaction. However, for the ABR results, although the authors added the control group to Figure 2, the control group was still excluded from the statistical analysis. Without statistically analyse the difference between control and other experimental groups, especially, between control and noise + drug groups, a piece of very important information is missing. That is whether or not the drug treatment would have any therapeutic value/potential, i.e., to make the hearing threshold return to the control level. Therefore, control group should be included in the analysis and the results need to be discussed accordingly. In addition, in lines 208 – 208, please clarify what does “the two-way repeated measures one-way analysis of variance (ANOVA)” mean?

7. PLOS authors have the option to publish the peer review history of their article (what does this mean?). If published, this will include your full peer review and any attached files.

Reviewer #1: **Yes: **Rachael Richardson

Reviewer #2: **Yes: **Yiwen Zheng

---

## [Author Response · Author response to Decision Letter 1]

19 Mar 2023

Rebuttal Letter

Manuscript ID: PONE-D-22-25555R1

Title of Manuscript: Round-window delivery of lithium chloride protects cochlear synapses from noise-induced excitotoxic trauma by inhibiting NMDA receptor

Authors’ Comment

We would like to thank the associate editor and the two reviewers for their valuable time and constructive comments. Below please find our response to the reviewers’ comments. Their comments are listed below followed by our responses summarizing how we addressed their concerns. We also included page numbers to indicate where changes were made in the manuscript. We hope that we have satisfactorily addressed the reviewers’ concerns. Due to insightful comments provided by the reviewers, the quality of our manuscript has been substantially improved. We want to thank the two reviewers and the editors for their valuable time and effort.

Reviewer #1: Thank you for the changes you made to the manuscript.

I have a couple of minor revisions that I think need to be addressed prior to publication.

Suggested change to the title from "Round-window delivery of lithium chloride regenerates rat cochlear synapses from noise-induced excitotoxic trauma by inhibiting NMDA receptor" to one of the following:

"Round-window delivery of lithium chloride regenerates cochlear synapses damaged by noise-induced excitotoxic trauma via inhibition of the NMDA receptor in the rat"

OR

"Regeneration of cochlear synapses in the rat following noise-induced excitotoxic trauma by round-window delivery of lithium chloride to inhibit the NMDA receptor" or something similar, because I don't think "regenerates rat cochlear synapses from noise-induced excitotoxic trauma" makes sense.

Author’s Response: Thank you for your comment. We changed the title to "Round-window delivery of lithium chloride regenerates cochlear synapses damaged by noise-induced excitotoxic trauma via inhibition of the NMDA receptor in the rat".

Line 208: "the two-way repeated measures one-way analysis of variance" Was it one way or two way?

Author’s Response: Thank you for your careful observation. There was typo. We used a two-way repeated measures analysis of variance (ANOVA).

Line 224 "in the noise + vehicle group had returned to that in the noise-only group." change to ...of the noise-only group.

Author’s Response: Thank you for your careful observation. We changed to “of”. 

Line 302 "There is a robust literature on potential therapeutic agent to prevent or regenerate noise-induced synaptic loss in the cochlea [7, 23-26]. " Change to "There is a robust literature on the delivery of potential therapeutic agents to prevent or regenerate noise-induced synaptic loss in the cochlea [7, 23-26].

Author’s Response: Thank you for your comment. We changed the sentence. 

Suggested change to Line 371 ""Fourthly, many key questions remain for clinical application. There is likely a significant clinical population with loss of cochlea synapse that could be addressed by a treatment analogous to that used here in the noise-exposed rat. Although the local delivery of lithium to the round window was effective in regenerating cochlear synapses 24 hours after the noise exposure, a key question would be how long this time window could be extended for effective synaptogenesis in a clinical setting."

Author’s Response: Thank you for your comment. We changed the last sentences as reviewer’s comment. 

Reviewer #2: The authors have addressed most of my comments to my satisfaction. However, for the ABR results, although the authors added the control group to Figure 2, the control group was still excluded from the statistical analysis. Without statistically analyse the difference between control and other experimental groups, especially, between control and noise + drug groups, a piece of very important information is missing. That is whether or not the drug treatment would have any therapeutic value/potential, i.e., to make the hearing threshold return to the control level. Therefore, control group should be included in the analysis and the results need to be discussed accordingly. In addition, in lines 208 – 208, please clarify what does “the two-way repeated measures one-way analysis of variance (ANOVA)” mean?

Author’s Response: We fully understand the reviewer’s concerns. The purpose of this study was to compare the protective effects on cochlea synapses after noise exposure. To achieve this, the control group should have similar variables to the experimental group except for not receiving lithium administration. Therefore, controls included animals exposed to noise who received only the vehicle (see abstract line 29-30). To avoid confusion, we have changed the term ‘control group’ to ‘normal group’ in the manuscript and figures. Additionally, we have corrected a typographical error in the statistical description in line 208 as follows: the two-way repeated measures analysis of variance (ANOVA).

---

## [Decision Letter · Decision Letter 2]

5 Apr 2023

Round-window delivery of lithium chloride regenerates cochlear synapses damaged by noise-induced excitotoxic trauma via inhibition of the NMDA receptor in the rat

PONE-D-22-25555R2

Dear Dr. Jung,

We’re pleased to inform you that your manuscript has been judged scientifically suitable for publication and will be formally accepted for publication once it meets all outstanding technical requirements.

Kind regards,

Alan G. Cheng, M.D.

Academic Editor

PLOS ONE

Additional Editor Comments (optional):

Reviewers' comments:

Reviewer's Responses to Questions

**Comments to the Author**

1. If the authors have adequately addressed your comments raised in a previous round of review and you feel that this manuscript is now acceptable for publication, you may indicate that here to bypass the “Comments to the Author” section, enter your conflict of interest statement in the “Confidential to Editor” section, and submit your "Accept" recommendation.

Reviewer #1: All comments have been addressed

Reviewer #2: All comments have been addressed

2. Is the manuscript technically sound, and do the data support the conclusions?

Reviewer #1: Yes

Reviewer #2: Yes

3. Has the statistical analysis been performed appropriately and rigorously? 

Reviewer #1: Yes

Reviewer #2: Yes

4. Have the authors made all data underlying the findings in their manuscript fully available?

Reviewer #1: Yes

Reviewer #2: Yes

5. Is the manuscript presented in an intelligible fashion and written in standard English?

Reviewer #1: Yes

Reviewer #2: Yes

6. Review Comments to the Author

Reviewer #1: (No Response)

Reviewer #2: (No Response)

7. PLOS authors have the option to publish the peer review history of their article (what does this mean?). If published, this will include your full peer review and any attached files.

Reviewer #1: **Yes: **Rachael Richardson

Reviewer #2: **Yes: **Yiwen Zheng

---

## [Editor Report · Acceptance letter]

11 May 2023

PONE-D-22-25555R2 

Round-window delivery of lithium chloride regenerates cochlear synapses damaged by noise-induced excitotoxic trauma via inhibition of the NMDA receptor in the rat 

Dear Dr. Jung:

I'm pleased to inform you that your manuscript has been deemed suitable for publication in PLOS ONE. Congratulations! Your manuscript is now with our production department. 

Kind regards, 

on behalf of

Dr. Alan G. Cheng 

Academic Editor

PLOS ONE